# *CSMD1* Mutations Are Associated with Increased Mutational Burden, Favorable Prognosis, and Anti-Tumor Immunity in Gastric Cancer

**DOI:** 10.3390/genes12111715

**Published:** 2021-10-28

**Authors:** Taobi Huang, Yuan Liang, Huiyun Zhang, Xia Chen, Hui Wei, Weiming Sun, Yuping Wang

**Affiliations:** 1The First Clinical Medical College, Lanzhou University, Lanzhou 730000, China; huangtb19@lzu.edu.cn (T.H.); liangy2018@lzu.edu.cn (Y.L.); zhy910825@163.com (H.Z.); xchen20@lzu.edu.cn (X.C.); weih20@lzu.edu.cn (H.W.); 2Department of Gastroenterology, The First Hospital of Lanzhou University, Lanzhou 730000, China; 3Key Laboratory for Gastrointestinal Diseases of Gansu Province, The First Hospital of Lanzhou University, Lanzhou 730000, China; 4Department of Endocrinology, The First Hospital of Lanzhou University, Lanzhou 730000, China; swm77@163.com

**Keywords:** *CSMD1*, TCGA, ICGC, gastric cancer, survival, tumor mutational burden, immune infiltration, immune checkpoint inhibitors, PD-L1

## Abstract

Tumor mutational burden (TMB) is considered a potential biomarker for predicting the response and effect of immune checkpoint inhibitors (ICIs). To find specific gene mutations related to TMB and the prognosis of patients, the frequently mutated genes in gastric cancer patients from TCGA and ICGC were obtained and the correlation between gene mutation, TMB, and prognosis was analyzed. Furthermore, to clarify whether specific gene mutations can be used as predictive biomarkers of ICIs, a gene set enrichment analysis (GSEA) for immune pathways and an immune infiltration analysis were conducted. The results showed that CUB and Sushi multiple domains 1 (*CSMD1*) mutation (*CSMD1*-mut) were associated with higher TMB and better prognosis in patients. The genetic map showed that, compared with wild-type samples, the loss of chromosomes 4q, 5q, 8p, and 9p decreased and the status of microsatellite instability increased in the *CSMD1*-mut samples. The GSEA analysis showed that immune-related pathways were enriched in the *CSMD1*-mut samples. The immune infiltration analysis showed that the anti-tumor immune cells were upregulated and that the tumor-promoting immune cells were downregulated in the *CSMD1*-mut samples. The gene co-expression analysis showed that PD-L1 expression was higher in the *CSMD1*-mut samples. In summary, *CSMD1*-mut in gastric cancer was associated with increased TMB and favorable survival and may have potential significance in predicting the efficacy of anti-PD-L1.

## 1. Introduction

According to the 2018 global cancer statistics and the 2015 Chinese cancer statistics, the number of new cases of gastric cancer was 1,033,701 (5.7%), ranking fifth in the world and second in China as the most common type of cancer. The number of deaths for gastric cancer was 782,685 (8.2%), making it the world’s third leading cause of cancer death [1,2]. Gastric cancer is the most common type of cancer that causes morbidity and mortality in some Asian countries. In China, the incidence and death toll of gastric cancer have been high, especially the mortality rate, which is higher than that of European and American countries and is related to the low rate of early diagnosis and early treatment of gastric cancer patients and the differences in the levels of diagnosis and treatment. The global burden of gastric cancer, especially in China, remains heavy. Endoscopic resection is the main treatment for early gastric cancer, but most cases are diagnosed as advanced gastric cancer and need surgery combined with first-line fluorouracil chemotherapy. Sequential chemotherapy based on platinum and fluoropyrimidine is the first choice for patients with metastatic gastric cancer [3]. In recent years, some molecular targeted drugs such as trastuzumab and ramucirumab (an antibody against vascular endothelial growth factor 2 receptor) have also been approved for the treatment of advanced gastric cancer, but the median survival rate of patients with gastric cancer (less than 1 year) is still very low [4]. There is an urgent need for new treatments to improve treatment and prognosis.

In recent years, immunotherapy is increasingly regarded as a potential innovative therapy in the field of cancer, in which new immune checkpoint inhibitors (ICIs) may become more important. With the development of next-generation sequencing (NGS), a large number of previously known or newly discovered genomic variants have been detected in the genome of gastric cancer [5,6,7]. Studies have found that gene mutations are associated not only with clinical features (such as age, sex, and tumor stage) but also with tumor mutational burden (TMB) [8]. On the one hand, mutations in driver genes can lead to oncogenesis. On the other hand, a large number of somatic mutations can produce new antigens. In this case, the higher TMB, the higher the mutation frequency of the cancer cells, so it is theoretically more likely to be effective for immunotherapy. Thus far, studies have confirmed that a high TMB can help predict the efficacy of tumor immunotherapy in patients with non-small-cell carcinoma, melanoma, and bladder cancer. As a potential biomarker of immunotherapy, TMB also shows better predictive results after anti-PD-1/PD-L1 (programmed death-ligand 1) immunotherapy in many other tumor types [9,10,11,12]. For some patients with chemotherapy-refractory gastric cancer with mismatch repair deficiency or microsatellite instability (MSI), the overall survival rate was also greatly improved after anti-PD-1/PD-L1 immunotherapy. However, this therapy is not effective for all patients with gastric cancer and new potentially accurate immunotherapy checkpoints are required. Some studies have shown that the median TMB of *TP53* mutant tumor samples is higher than that of wild-type tumor samples and that patients with high TMB may respond to immune checkpoint inhibitors [8]. However, it is not completely clear whether the TMB of the gastric cancer genome is related to the prognosis and immunotherapy response of patients.

In this study, The Cancer Genome Atlas (TCGA) dataset and International Cancer Genome Consortium (ICGC) dataset were used to mine genes with frequent mutations in gastric cancer cohorts via bioinformatics methods and to explore the correlation between these gene mutations, TMB, and the prognosis of patients with gastric cancer. Furthermore, an immune-related pathway enrichment analysis, an immune infiltration analysis, and an immune checkpoint molecular expression were carried out to explore the relationship between gene mutation and immune function. The aim of this study was to find potential biomarkers that could predict TMB, prognosis, and the efficacy of ICIs in gastric cancer.

## 2. Materials and Methods

### 2.1. Data Sources and Visualization

The somatic mutation data and corresponding clinical data of stomach adenocarcinoma (STAD) were downloaded from the TCGA database (https://www.cancer.gov accessed on 20 September 2020), and maf files were visualized using maftools package. The somatic mutation data and corresponding clinical data of gastric cancer in China (GACA-CN) were downloaded from the ICGC database (https://icgc.org accessed on 13 October 2020). After annotated, tsv files were visualized with GenVisR package. Only mutated genes that cause changes in amino acids were included. The total samples included 437 patients in TCGA-STAD cohort and 120 patients in GACA-CN cohort. Considering that the largest sample size was needed to obtain the most real results, we only used the TCGA cohort data for analysis in the later stage.

### 2.2. TMB Value Calculation and Survival Analysis

TMB is the total number of mutations per Mb in tumor tissue (excluding synonymous mutations), including the total number of gene coding errors, base substitutions, and gene insertions or deletions. According to the mutation condition, the samples were divided into the wild-type and mutation groups, and the TMB of the two groups was calculated as the number of non-synonymous protein coding variants divided by the total sequencing exon length (38 Mb was used as an estimate of exon size). Gene mutations that did not cause amino acid changes were not counted. The estimated TMB of each sample was the total number of gene mutations per 38 Mb. The results were visualized by a violin plot, and the correlation between gene mutations and TMB was evaluated by a *t*-test. In order to further evaluate the relationship between gene mutations and clinical prognosis, we performed Kaplan–Meier analysis with survival package. *p* value < 0.05 was considered to have a statistically significant difference.

### 2.3. Immunologic Gene Set Enrichment Analysis

GSEA V4.1.0 software (Boston, Massachusetts, US)was used to conduct a gene set enrichment analysis and is a tool developed by the Broad Institute to correlate gene sets with the entire transcriptome expression profile for a function-specific gene set [13]. The expression dataset of the gastric cancer samples (genome of reference: hg38) was downloaded from the TCGA database and divided into the wild-type and mutation groups according to their CUB and Sushi multiple domains 1 (*CSMD1*) mutation (*CSMD1*-mut) status. The immunologic gene sets were downloaded from the official website of GSEA (https://www.gsea-msigdb.org/gsea/msigdb/collections.jsp#C2 accessed on 9 November 2020). In order to get a more accurate *p* value, 1000 permutations were performed by the analysis. The pathways with |normalized enrichment score (NES)| > 1, nominal *p* value < 0.05 and false discovery rate (FDR) *q*-value < 0.25 were considered to be significantly enriched.

### 2.4. Immune Infiltration Analyses

CIBERSORT is a deconvolution algorithm that can evaluate the infiltration abundance of immune cells in a sample [14]. We downloaded the composition of 22 kinds of immune cells of gastric cancer samples from TCGA website (https://gdc.cancer.gov/about-data/publications/panimmune accessed on 28 October 2020) and the ggplot2 package and corrplot package were used to visualize the proportion distribution of immune infiltration in each sample and the correlation between immune cells. In the correlation matrix, red indicated positive correlation, blue indicated negative correlation, and the larger the absolute value, the stronger the correlation. TIMER2.0 (http://timer.cistrome.org accessed on 25 September 2021), using six different algorithms based on immune deconvolution, including TIMER, xCell, MCP-counter, CIBERSORT, EPIC and quantTIseq, was used to analyze the relationship between gene expression, mutation status, copy number variation, clinicopathological features and immune cell infiltration in TCGA database [15]. “Mutation module” was used to analyze the correlation between *CSMD1*-mut and immune infiltration. Log2 Fold Change > 0 indicated a higher level of infiltrating cells in the mutational samples, while log2 Fold Change < 0 indicated a lower level. Wilcoxon test was used for statistical analysis, and *p*-value < 0.05 indicated that there was statistical difference.

### 2.5. Gene Co-Expression, Genomic Alterations and Pathway Analyses

The “co-expression” module in cBioportal (http://www.cbioportal.org accessed on 26 September 2021) was used to identify genes co-mutated or co-expressed with *CSMD1*, and the mRNA expression data were quantified by RSEM (Batch normalized from Illumina HiSeq_RNASeqV2). The “clinical” module was used to identify biogenetics alterations associated with *CSMD1*-mut. The “pathway” module presented the biological pathways of co-expressed genes related to *CSMD1*-mut.

## 3. Results

### 3.1. Identification of Co-Mutated Genes in TCGA and ICGC Gastric Cancer Cohorts

There were 17,431 mutated genes and a total of 142,177 somatic mutations in the TCGA-STAD cohort. The average number of mutations in each sample was 325, ranging from 1 to 5929. Missense mutations were the largest variant classification. The incidence of transition (C > T) is still significantly higher than that of other single nucleotide variants (SNV) classifications (Appendix A). The waterfall showed the top 30 genes with frequent mutations and genes altered in 414 (94.74%) of the 437 samples. Among them, *TTN* (53%), *TP53* (46%), and *MUC16* (32%) were the three most frequently mutated genes (Figure 1A). There were 30,116 somatic mutations found in 6315 genes causing amino acid changes in the GACA-CN cohort. The waterfall showed the top 30 genes with frequent mutations, in which *TP53* and *TTN* were the two most frequently mutated genes (Figure 1B). In order to obtain genes mutated in both the GACA-CN and TCGA-STAD cohorts, a Venn diagram was used to show nine commonly mutated genes in both cohorts including *TP53*, *TTN*, *CSMD1*, *SYNE1*, *CSMD3*, *OBSCN*, *ZFHX4*, *PLEC*, and *COL12A1* (Figure 1C).

### 3.2. CSMD1-Mut Was Associated with TMB and Survival

The TMB of the samples with *TTN*, *CSMD1*, *SYNE1*, *CSMD3*, *OBSCN*, *ZFHX4*, *PLEC*, and *COL12A1* mutations was significantly higher than that of the wild-type samples in the TCGA cohort (*p* < 0.05). To explore the correlation between high TMB and survival in patients with these mutations, we performed a Kaplan–Meier analysis. The results showed that only *CSMD1*-mut was associated with patient survival and that patients with *CSMD1*-mut had a better prognosis than those with *CSMD1*-wild (*p* = 0.028 < 0.05) (Figure 2A,B).

### 3.3. Profiles of CSMD1-Mut and Functional Domains

We further analyzed the number, sites, types, and domains of *CSMD1*-mut. There was a total of 111 *CSMD1*-muts among 79 of the 437 (18.08%) samples including 102 missense mutations, 3 nonsense mutations, 2 frame shift insertions, 2 frame shift deletions, and 1 splice site and 7 sites with 2 or more mutations in the TCGA cohort. Interestingly, there were multiple mutations in the PHA02927 domain (secreted complement-binding protein), which was the most common structure found in many complementary inhibitors containing such multiple domains [16] (Figure 3A). It was not known whether mutations in this region were directly related to patient survival, but it aroused our interest in the continued exploration of whether *CSMD1*-mut is associated with immune-related molecules, infiltrating cells, and signaling pathways.

### 3.4. Characteristic Alterations of Chromosomes and Genome

To assess the biogenetic impact of CSMD1-mut, the gain and loss of chromosomes, and the status of microsatellite instability (MSI) were assessed and genome alterations analyses were performed. Compared with the wild-type samples, the loss of chromosomes 8p, 4q, 9p, 5p, and 4p were reduced in the CSMD1-mut samples (Figure 3B). What was more important was the molecular characteristics of the gastric cancer subtypes. In patients with CSMD1-mut, the proportion of genomically stable (GS) and chromosome instability (CIN) subtypes decreased significantly while MSI increased significantly, and the sample proportion of polymerase epsilon (POLE) mutation also increased (Figure 3C,D). The statistical methods and other, more detailed data are listed in Appendix A. The immune signaling pathways were enriched in the CSMD1-mut samples.

To explore whether immunological pathways play a role in the correlation between *CSMD1*-mut and prognosis, we performed a gene set enrichment analysis. As shown by Figure 4A–H, the gene sets related to the immune signaling pathways were significantly enriched in the *CSMD1*-mut group, including C-type lectin receptors (CLRs), downstream signaling events of B cell receptor (BCR), three pathways related to inlerleukin-12 (IL-12) (protein expression by JAK-STAT signaling after interleukin 12 stimulation, interleukin 12 signaling, and interleukin 12 family signaling), and three pathways related to antigen presentation (presentation of soluble exogenous antigens endosomes, antigen processing cross presentation, and HDAC in antigen presentation down). These results suggested that the immune-related signaling pathways were significantly upregulated in the *CSMD1*-mut samples.

### 3.5. Immune Infiltration in the Tumor Microenvironment of CSMD1-Mut

The proportions of 22 kinds of immune cells in the 437 gastric cancer samples are shown in Figure 5A. The correlation matrix between the immune cells showed that the strongest positive correlation was found between neutrophils and activated mast cells, that the strongest negative correlation was found between resting NK cells and activated NK cells, and that negative correlations were found between resting memory CD4^+^ T cells and CD8^+^ T cells and between M2 macrophages and naïve B cells (Figure 5B). Anti-tumor immune cells including CD4^+^ Th1 cells (*p* = 0.009), NK cells (*p* = 0.033), M1 macrophage cells (*p* = 0.007), and plasmacytoid dendritic cells (PDC) (*p* = 0.010) had higher proportions in the *CSMD1*-mut group (Figure 5C,F,G,I). Tumor-promoting immune cells including regulatory T (Treg) cells (*p* = 0.007), M2 macrophage cells (*p* = 0.037), and endothelial cells (*p* = 0.007) had lower proportions in the *CSMD1*-mut group (Figure 5D,H,J). B cells had a lower proportion in the *CSMD1*-mut group (Figure 5E). There was a higher trend of CD8^+^ T cells in the mutation group, although there was no statistical difference between the mutational and wild-type groups (Appendix A). 

### 3.6. Analysis of Co-Expression Genes and Signaling Pathways Related to CSMD1-Mut

In order to further understand whether *CSMD1*-mut was related to the expression of immune checkpoints, co-expression analyses were carried out. The results showed that there was a significant positive correlation between *CSMD1*-mut and the expression of *CD274* (PD-L1; *p* = 5.506 × 10^−3^), and a tendency of higher expression of *PDCD1* (PD-1), *CTLA4*, *LAG3*, *TIGIT* and *HAVCR2* (TIM3) in *CSMD1*-mut group (Figure 6; Appendix A). Signaling pathway analyses showed that there were ten related pathways, in which RTK-RAS, HIPPO and WNT pathways were the three most relevant signaling pathways with scores of 35, 26 and 25, respectively (Appendix A; Appendix A). Furthermore, the correlation between the expression of genes involved in the related pathways and *CSMD1*-mut was analyzed, and the results showed that the expression of *TP53* and *ALK* was positively correlated with *CSMD1*-mut.

## 4. Discussion

In this study, in order to obtain frequently mutant genes in gastric cancer cohorts, we systematically analyzed the data and rules of somatic mutations in the TCGA and ICGC databases and obtained the top 30 genes with frequent mutations in the two cohorts. Interestingly, only nine genes were mutated in both cohorts. Most of the TCGA-STAD cohort were Western patients, while the patients in the GACA-CN cohort were Chinese. Inherent differences between different races may play a role. The result may also be caused by inconsistencies in the group entry criteria, the sample collection process, the sample storage methods, and the sequencing methods. Since TMB can predict the survival of multiple tumor types after immunotherapy, we then analyzed the correlation between these nine gene mutations, TMB, and survival [17]. The results showed that patients with *CSMD1*-mut had significantly increased TMB and better prognoses than *CSMD1*-wild patients.

*CSMD1*, located on the short arm of human chromosome 8 (8p23.2), is mainly expressed in the central nervous system and epithelial tissue. *CSMD1* has been proven to be a tumor suppressor gene in multiple tumors and has copy number variations, somatic mutations, deletions, aberrant splicing, and chromosome aberrations in leukemia, primary lung cancer, head and neck squamous cell carcinoma, breast cancer, liver cancer, and other cancers [18,19,20,21]. The role of *CSMD1* in gastric cancer has not been studied until the past two years. Chen et al. confirmed that *CSMD1* can be downregulated by microRNA-10b to promote the growth, proliferation, and invasion of gastric cancer cells, which was the first and only evidence of the role of *CSMD1* in gastric cancer [22]. However, so far, there is no research to explain the general situation of *CSMD1*-mut in gastric cancer and the alterations in biological function caused by it. Interestingly, our analysis results show that the prognosis of *CSMD1*-mut samples is better than that of wild-type samples. One possible explanation for this result is that functional site mutations in *CSMD1*-mut patients have a positive effect on the prognosis of patients with gastric cancer. Another possible explanation is consistent with the results of other similar studies: that high TMB samples are more likely to benefit from immunotherapy to achieve a better prognosis [8]. Recent studies have confirmed that, in addition to MSI, EBV, and PD-L1, TMB can also be used as a biomarker to predict the prognosis of patients with advanced gastric cancer who receive immune checkpoint inhibitors [23]. We prefer a combination of two factors: most likely *CSMD1*-mut not only increased the TMB of the samples but also affected the functional domain of the *CSMD1* protein or immune-related signaling pathways. Therefore, we visualized the mutation sites and domains of *CSMD1* and found that multiple mutations occurred in the secreted complement-binding protein domain. Subsequently, we found that several immune-related signaling pathways were upregulated in the *CSMD1*-mut samples; that there was a higher proportion of anti-tumor immune cells including CD4^+^ Th1 cells, NK cells, M1 macrophage cells and PDC; that there was a lower proportion of tumor-promoting immune cells, including Treg cells, M2 macrophage cells, and endothelial cells; and that there was upregulation of PD-L1. We speculated that there may be a relationship between mutations of the PHA02927 domain in the *CSMD1*-mut samples and the enrichment of the above immune signaling pathways, the infiltration of anti-tumor immune cells, and the upregulation of PD-L1, which was related to the relatively good prognosis of patients with gastric cancer. In addition, Treg cells not only play a suppressive role in anti-tumor immunity but also may participate in anti-PD-1/PD-L1 resistance mechanisms [24,25]. Patients with *CSMD1*-mut had less Treg cells and higher PD-L1 expression, which helped them benefit more from anti-PD-L1 and to less likely develop drug resistance.

The increase in mutation rate is a characteristic change of human cancer. Exploring the rules of somatic mutations in gastric cancer is very important in exploring the mechanism of occurrence and development of gastric cancer. According to the comprehensive molecular evaluation of 295 cases of primary gastric adenocarcinoma, the TCGA project divided gastric cancers into four molecular subtypes including Epstein–Barr virus (EBV)-infected tumors, MSI tumors, GS tumors, and CIN tumors [26]. These four molecular subtypes can predict prognosis and can guide treatment. The prognosis of EBV tumors is the best, and that of GS tumors is the worst. CIN patients benefit most from adjuvant chemotherapy, while GS patients benefit least from adjuvant chemotherapy [27]. Patients with a high MSI can hardly benefit from adjuvant chemotherapy but are excellent candidates for immunotherapy [28]. Having fewer GS tumors in *CSMD1*-mut patients predicted better prognoses, while having fewer CIN tumors indicated that fewer people can benefit from adjuvant chemotherapy, but the significant increase in MIS among *CSMD1*-mut patients meant that these patients were more suitable for immunotherapy. The *POLE* mutation was an independent indicator for predicting survival benefits from immunotherapy, independent from MSI [29], which is also listed as one of the molecular subtypes of gastric cancer in the cBioportal database. Interestingly, *CSMD1*-mut patients had more *POLE* mutations. These molecular subtypes showed obvious characteristics of genomic alterations. Previous studies on chromosomal aberration maps in tumors showed that the deletion or under-expression of chromosomes often occurs in most solid tumors, which leads to tumor progression [30,31]. The loss of these chromosomes was reduced in *CSMD1*-mut patients, and we speculate that these changes may help to resist or delay the progression of gastric cancer.

However, our study also has some limitations. First, our subsequent analyses are based on the data from the TCGA cohort, and whether these results are applicable to the Chinese patients with gastric cancer needs to be further verified. Second, these results are only based on the cohort follow-up survey from a public database, and the effect of *CSMD1*-mut on the growth and metastasis of gastric cancer needs to be verified by experiments in vivo and in vitro. Finally, whether *CSMD1*-mut can be used as a predictive biomarker for the efficacy of anti-PD-L1 in patients with gastric cancer and even other tumors remains to be further verified and practiced.

## 5. Conclusions

In summary, *CSMD1*-mut in gastric cancer was associated with increased TMB and favorable survival. Significant features, including upregulation of the immune pathway, enrichment of the anti-tumor immune cells in a tumor microenvironment, the reduction in tumor-promoting immune cells, a high TMB, a high MSI status, and the upregulation of PD-L1, indicated that gastric cancer patients with *CSMD1*-mut may benefit from anti-PD-L1. This study provides a new idea for immunotherapy and for biomarkers as predictors of the immune response in gastric cancer, and more prognostic information and opportunities for immunotherapy in patients with gastric cancer.

## Figures and Tables

**Figure 1 genes-12-01715-f001:**
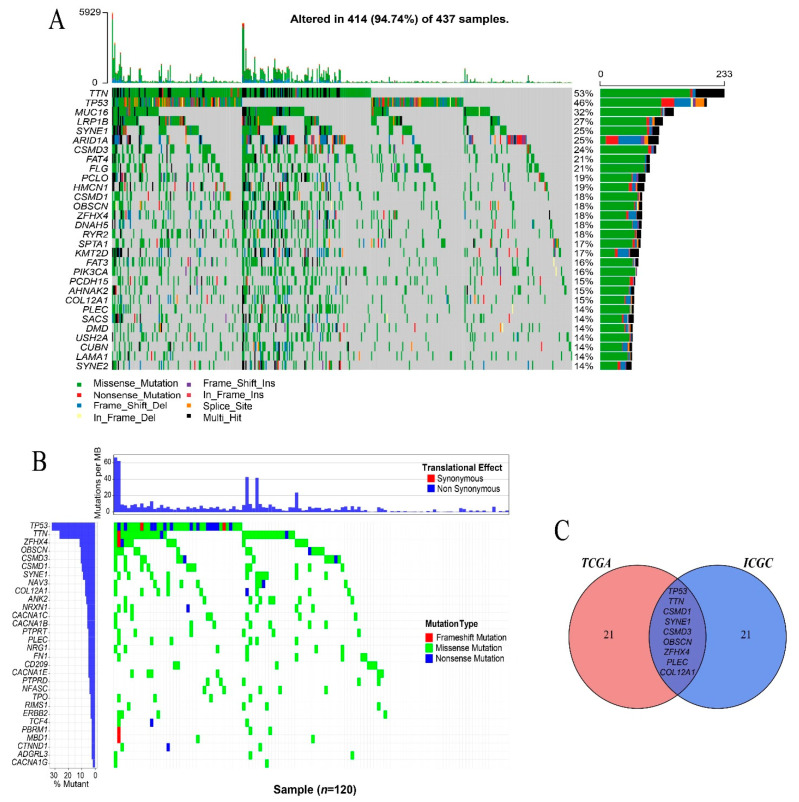
Gene mutation landscape in gastric cancer. (**A**) The waterfall plot of top 30 genes with frequent mutations in the TCGA gastric cancer cohort shows the names of the top 30 genes on the left, the frequency of gene mutation on the right, the total number of somatic mutations in each sample at the top, and the type of mutation at the bottom. (**B**) The waterfall plot of the top 30 genes with frequent mutations in the ICGC gastric cancer cohort shows the name and mutation frequency of the top 30 genes on the left, the average number of mutations per Mb in each sample at the top, and the type of mutation on the right. (**C**) The Venn diagram shows nine genes mutated in both of the TCGA and ICGC cohorts.

**Figure 2 genes-12-01715-f002:**
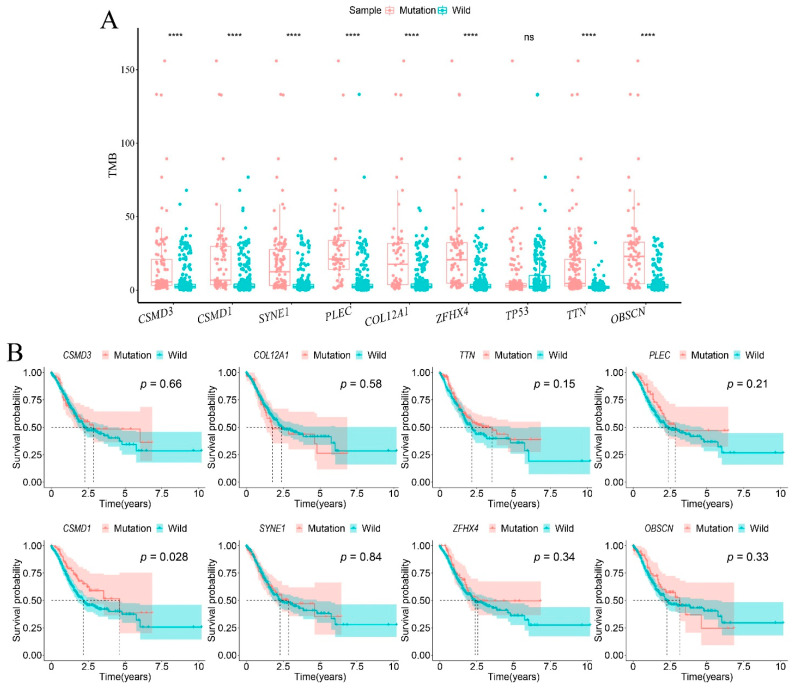
The correlation between gene mutation, TMB, and survival. (**A**) The box plot shows that all of the other eight gene mutations except *TP53*, were associated with TMB. In the *t*-test, **** *p* < 0.0001; ns, *p* >= 0.05. (**B**) The survival curve showed that only *CSMD1*-mut is associated with survival time in patients. The *p*-value in the log-rank test is marked in each diagram.

**Figure 3 genes-12-01715-f003:**
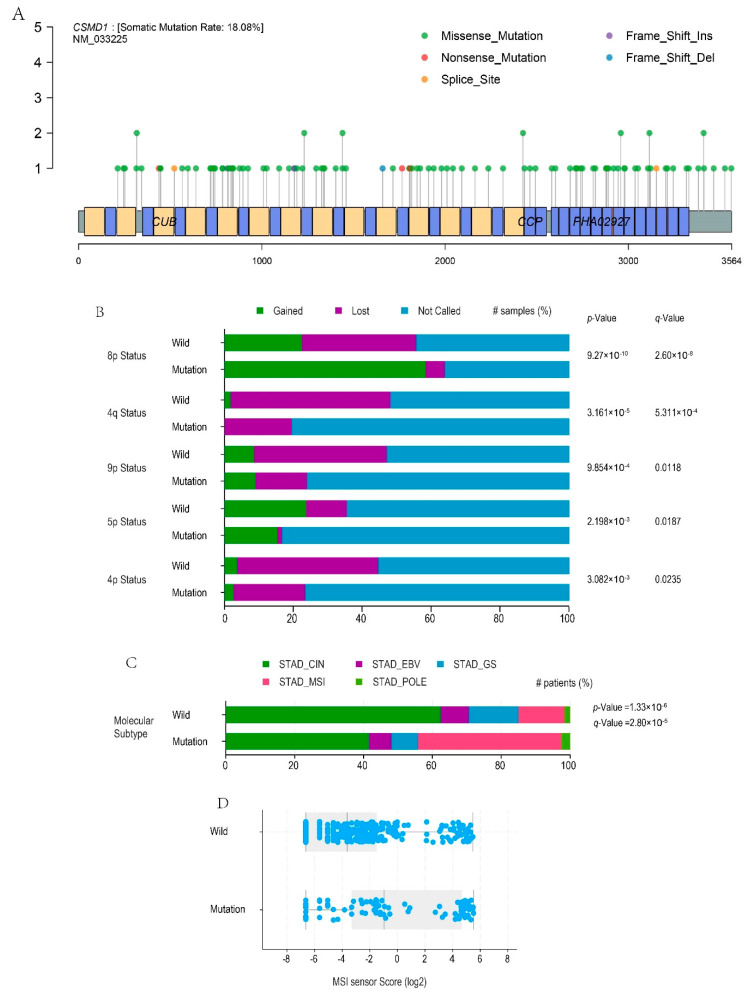
*CSMD1*-mut profile and biogenetic alterations ((**B**–**D**) were from yjr cBioportal website). (**A**) The location, type, and number of *CSMD1*-mut and their relationship with known domains. Blue indicates the CCP domain, and yellow indicates the CUB domain. The number of mutations is on the left, and the types of mutations are at the bottom. *CSMD1* contains four members of the PHA02927 super family, also known as the secreted complement binding protein, which is composed of multiple CCP conservative domains in series. (**B**) The statuses of chromosomes 8p, 4q, 9p, 5p, and 4p in the chi-squared test: green for gained, purple for lost, and blue for not called. (**C**) Molecular subtype analyses in the chi-squared test. (**D**) MSI status in the Wilcoxon test.

**Figure 4 genes-12-01715-f004:**
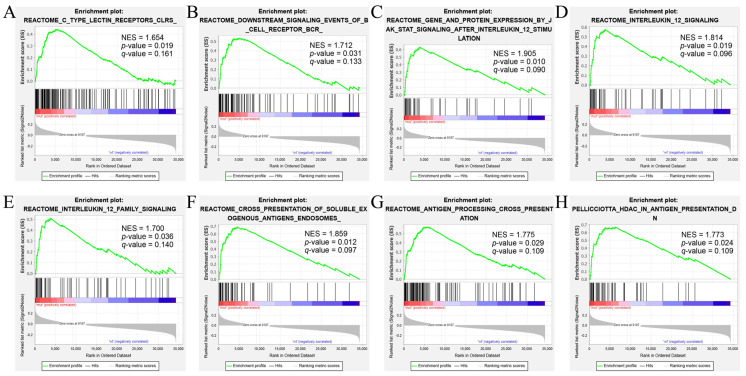
Immune-related pathways enriched in CSMD1-mut samples using GSEA program. (**A**) Reactome C-type lectin receptors (CLRs), (**B**) Reactome downstream signaling events of B cell receptor (BCR), (**C**) Reactome protein expression by JAK-STAT signaling after interleukin 12 stimulation, (**D**) Reactome interleukin 12 signaling, (**E**) Reactome interleukin 12 family signaling, (**F**) Reactome presentation of soluble exogenous antigens endosomes, (**G**) Reactome antigen processing cross presentation, and (**H**) HDAC in antigen presentation down. NES, Normalized enrichment score. P-value has been marked in each diagram.

**Figure 5 genes-12-01715-f005:**
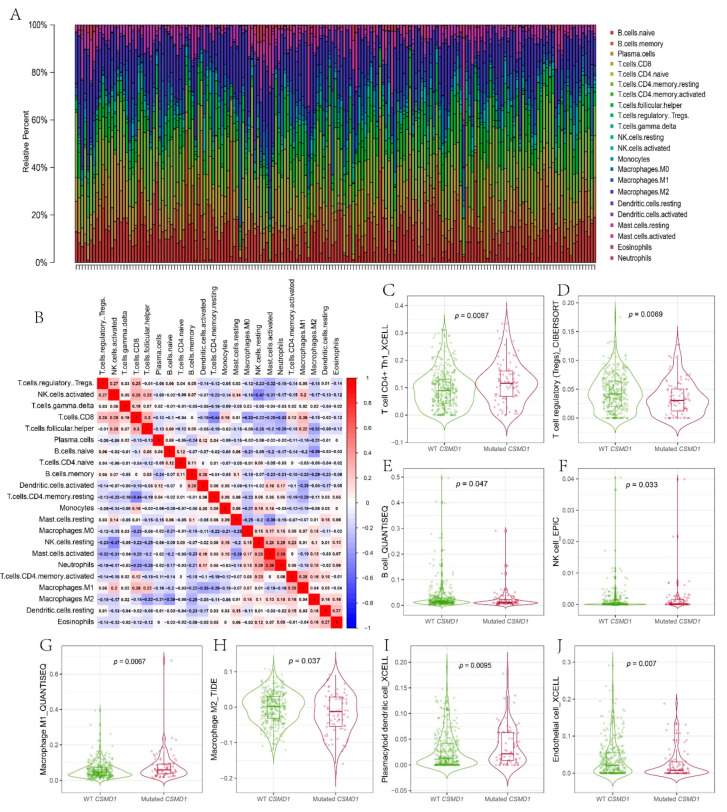
Immune infiltration. (**A**) The proportion of 22 kinds of immune cells in 437 gastric cancer samples. (**B**) The matrix of correlation between immune cells. Infiltration associated with anti-tumor immunity; (**C**) CD4+ Th1 cells, (**F**) NK cells, (**G**) M1 macrophage cells and (**I**) PDC. Infiltration associated with promoting tumor immunity (**D**) Treg cells, (**H**) M2 macrophage cells and (**J**) endothelial cells. (**E**) B cells. Figure 5C–J were from TIMER2.0 website. Wilcoxon test for statistics.

**Figure 6 genes-12-01715-f006:**
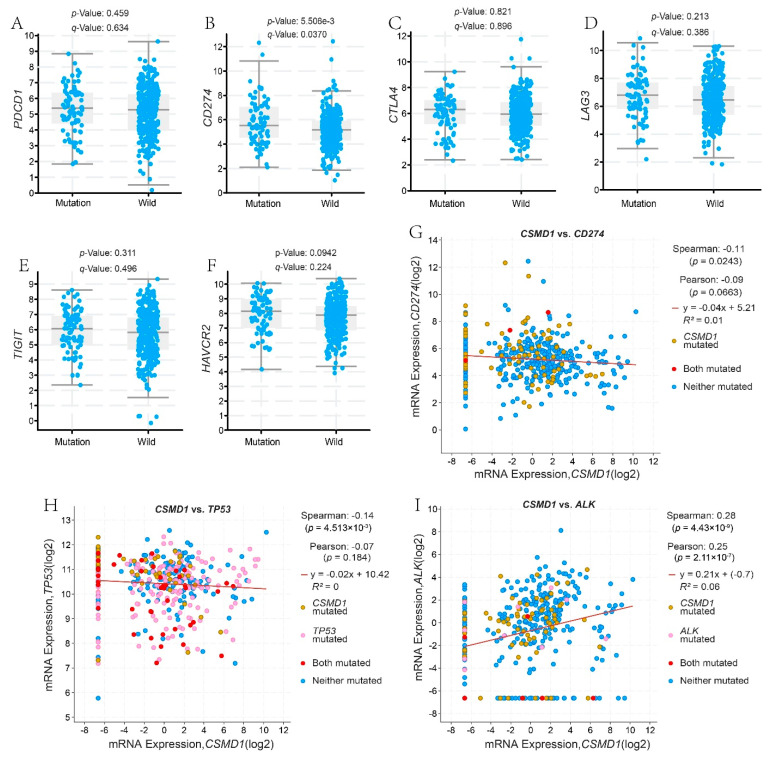
Correlation between expression of immune checkpoint molecules and *CSMD1*-mut; (**A**) PD-1 (*PDCD1*), (**B**) PD-L1 (*CD274*), (**C**) *CTLA4*, (**D**) *LAG3*, (**E**) *TIGIT* and (**F**) TIM3 (*HAVCR2*). (**G**) Co-expression of *CSMD1* and PD-L1 (*PDCD1*). Correlation between the expression of genes involved in related signaling pathways and *CSMD1*-mut; (**H**) *TP53* and (**I**) *ALK*. The figures were from cBioportal website.

## Data Availability

All of the data in the study are included in the article/Appendix A. Further inquiries can be directed to the corresponding authors.

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
