# Peer review of "CSMD1 Mutations Are Associated with Increased Mutational Burden, Favorable Prognosis, and Anti-Tumor Immunity in Gastric Cancer"

_genes, 2021, doi:10.3390/genes12111715_

Round 1
Reviewer 1 Report
Huang et al. investigated the association between CSMD1 mutations and mutational burden in gastric cancer and their potential utility as predictors of therapeutic efficacy of immunotherapy. The authors mined publicly available datasets of gastric cancer on TCGA and ICGC databases to identify the most commonly mutated genes. Of these, they focused their analysis on CSMD1 mutations that correlated with a better prognosis in gastric cancer. They found that CSMD1-mutated gastric cancer samples displayed enrichment of immune signaling pathways. Overall, the paper provides a correlative analysis of CSMD1 mutations and the associated genetic and prognostic changes in gastric cancer. I believe the paper would benefit from these comments:
Major comments
- The title does not reflect the findings of the study. The paper did not analyze the correlation between CSMD1 mutations and response to PD-1/PD-L1 immune checkpoint inhibitors, and the analysis was confined to the prognosis regardless of the therapeutic modality. Further analysis will be needed to confirm that these mutations could serve as predictors of response. Thus, I suggest changing the title to:
“CSMD1 mutations are associated with increased mutational burden, favorable prognosis, and enrichment of immune signaling pathways in gastric cancer”
- The abstract needs significant changes: I would suggest removing “R software”, “TIMER2.0” and “CBioportal” and focusing on common terminology of methodological approaches such as GSEA. These technicalities can be detailed in the methods. The phrase “The genetic map showed that 22 the loss of chromosome 4, 5q, 8p and 9p in CSMD1-mut samples decreased but the gain increased” is not very clear, and perhaps rephrasing is needed for clarity.
- I would suggest avoiding the terms “American and Chinese samples” in the abstract and throughout the manuscript. Samples in TCGA and ICGC databases are not necessarily American or Chinese. Thus, “samples from TCGA and ICGC databases” is more appropriate. For example Line 85: American gastric cancer; Line 88: Chinese gastric cancer; Line 144: American cohort; Line 155: Chinese and American cohorts; Also Line 266.
- I would suggest adding the statistical tests employed to all figure legends
- In Figure 1C, 70% (21/30) of mutations were not shared between gastric cancer samples from TCGA and ICGC databases, which is a large percentage. The authors need to comment on this disparity between data on the same malignancy from the 2 databases.
- The authors analyzed the correlation between CSMD1 mutations and prognosis (assessed by survival) not the therapeutic response to immune checkpoint inhibitors. I would suggest the authors make it more clear throughout the manuscript to avoid confusion.
- In lines 285-287 “Interestingly, our analysis results show that the 284 prognosis of CSMD1-mut samples is better than that of wild samples. One possible explanation for this result is that CSMD1-mut has a positive effect on the prognosis of patients 286 with gastric cancer”. This explanation is quite redundant and does not add much. A better explanation is needed.
Minor comments:
- Several typos need to be changed: e.g. Line 36: “deaths cases”: delete cases; Line 58: driver genes; Line 194, “the loss of chromosome 8p, 4q, 9p, 5p, 4p reduced in CSMD1-mut samples”. “Signal pathways” need to be changed to “signaling pathways” throughout the manuscript
- CSMD1 and other gene mutations (e.g. TP53) need to be italicized throughout the manuscript
- “Wild” needs to be changed to “wild-type” throughout the manuscript.
- Line 198: POLE needs to be spelled out in Section 3.4 and legend of Fig. 3. The rationale of investigating POLE is needed in Section 3.4
- In line 324, “finding the rules of somatic mutations in gastric cancer” is not very clear.
Author Response
Dear reviewer 1,
Thank you for commenting on my paper. Here is my reply. Please see the attachment.

Reviewer 2 Report
In the manuscript titled CSMD1 mutation is associated with tumor mutational burden and prognosis, and can predict therapy efficacy of PD-L1 in gastric cancer, Huang T. et al. using available databases and enrichment analysis, describe the predicting effects on the impact on the effectiveness of immunotherapy in gastric cancer.
The analyzes presented by Huang T. et al. are of great scientific importance, because of the difficulties in gastric cancer treating.
The manuscript presented for review is written very concisely and logically, however requires a few corrections:
- There is no information about the CSMD1 gene in the introduction. It is not known why this particular gene was analyzed.
- All gene names should be in italics.
- Figures in the supplement should be described: what they refer to, what they represent, etc. The same as the other figures in the text.
- In figure 1C it would be good to name at least these 9 genes mutated in both TCGA and ICGC cohorts.
- Figure A does not indicate that CSMD1-mut patients have a better prognosis or survival. Please correct the description of the Figure or change the text.
- Figure 3 should be under subsection 3.3
- Opening the S1 table is difficult due to some errors. Additionally, it is necessary to describe the table (what it applies to) in the file and in the text. The table is not clear after reading the text of the manuscript.
- If the figures contain images from websites or statistical programs, this information should be included in the description of the figures.
- Table S2 should contain short description.
- Table S3 should contain a short description. The text mentions TIM3 gene which is not included in the table, while the table includes genes such as CD274 which are not mentioned in the text. A more detailed description would be useful.
- Figure S2 should contain short description.
- The sentence "One possible explanation ... (line285-287) does not fully explain the fact that “the prognosis of CSMD1-mut samples is better than that of wild samples”. Please try to describe it more clearly.
- line 53: after the "immune checkpoint inhibitors", please add the abbreviation ICI (because this name appears in the text for the first time)
- line 63: NSCLC – the abbreviation should be expanded
- line 103: t-test instead of “T-test”
- lack of spaces:
- lines 105, 116, 117, 234, 235, 236, 238 – p value
- line 150 - between genes and percentages
- line 185 – between “domain” and “(secreted….
- line 110: ref. 13 is superscript
- line 112: CSMD1 - the abbreviation should be expanded
- line 304, 306: Treg instead of “treg”
- line 197: instability instead of “Instability”
Author Response
Dear reviewer 2,
It's a great pleasure for us to gained the recognition of you. Here is my reply. Please see the attachment.

Reviewer 3 Report
Could you please clarify the association between this mutation and the severity degree of the disease? And with the mutation rate?
How do you plan to do this in animal models? Could you clarify?
In my point of view, there's a need to add a take-home message regarding the CSMD1 mutation and the prediction of the disease.
Author Response
Dear reviewer 3,
It's a great pleasure for us to gained the recognition of you. Here is my reply. Please see the attachment.
